



# 1   A theory of glacial cycles: resolving Pleistocene puzzles

Hsien-Wang Ou[1]
[1] Department of Earth and Environmental Sciences, Lamont-Doherty Earth Observatory of Co-
lumbia University, Palisades, NY10964, USA (retired)
*Correspondence to*: Hsien-Wang Ou (hsienou0905@gmail.com)
**Abstract.** Since the summer surface air temperature that regulates the ice margin is anchored on
the sea surface temperature, we posit that the climate system constitutes the intermediary of the
orbital forcing of the glacial cycles. As such, the relevant forcing is the annual solar flux ab-
sorbed by the ocean, which naturally filters out the precession effect in early Pleistocene but
mimics the Milankovitch insolation in late Pleistocene. For a coupled climate system that is
inherent turbulent, we show that the ocean may be bistable with a cold state defined by the freez-
ing point subpolar water, which would translate to ice bistates between a polar ice cap and an ice
sheet extending to mid-latitudes, enabling large ice-volume signal regardless the forcing ampli-
tude so long as the bistable thresholds are crossed. Such thresholds are set by the global convec-
tive flux, which would be lowered during the Pleistocene cooling, whose interplay with the ice-
albedo feedback leads to transitions of the ice signal from that dominated by obliquity to the
emerging precession cycles to the ice-age cycles paced by eccentricity. Through a single dy-
namical framework, the theory thus may resolve many long-standing puzzles of the glacial cy-
cles.
**1   Introduction**



In late Pleistocene, the global ice volume exhibits pronounced variation at orbital fre-
quencies (Hays et al. 1976). As Antarctica is largely iced over since about ten million years
ago (Berger 1979), the ice signal reflects mainly that of the northern ice sheet, whose correla-
tion with the Milankovitch insolation supports the latter's control of the ice volume (Milan-
kovitch 1941). The linkage however is necessarily nonlinear since the 100-ky eccentricity con-
tains little power, yet it dominates the ice-age cycles of the late Pleistocene, the so-called "100-
ky problem" (Elkibbi and Rial 2001). Equally puzzling, the ice signal exhibits mainly the
obliquity periodicity in the early Pleistocene despite the greater precession amplitude (Raymo
and Nisancioglu 2003), and then the mid-Pleistocene transition (MPT, all acronyms are listed
in Appendix A) from the obliquity- to the eccentricity-dominated ice cycles is not accompanied
by appreciable change in the Milankovitch insolation (Clark et al. 2006). Given the distinct-
ness of these features, they should emerge from fundamental physics of the climate-cryosphere
system, which is yet to be delineated.
The above observations have weeded out some previous resolutions of the 100-ky prob-
lem, as briefly recounted below. Early attempts have invoked internal oscillations of the ice
sheet due to mass-balance feedback or isostatic adjustment (Weertman 1976; Imbrie and Im-
brie 1980; Oerlemans 1982; Pollard 1983; Pelletier 2003), but these are broadband processes,
which would elevate the low-frequency variance but not produce a spectral peak coincidental
with the eccentricity (Imbrie et al. 1993). Others have invoked stochastic or chaotic dynamics
in conjunction with the orbital forcing (Wunsch 2003; Huybers 2009), which however may be
at odds with the observation that ice ages seem to always terminate at rising eccentricity

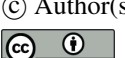



(Raymo 1997; Kawamura et al. 2007; Lisiecki 2010), suggesting a more deterministic process
(Meyers and Hinnov 2010). Both internal oscillations and stochastic dynamics may not ex-
plain why they are not operative in early Pleistocene when the northern hemisphere glaciation
has already set in (Ravelo et al. 2004) and yet 100-ky cycles are absent. Then there are con-
ceptual or dynamical-system models (Saltzman et al. 1984; Ghil 1994; Paillard 1998;
Tziperman et al. 2006; Imbrie et al. 2011; Crucifix 2013; Daruka and Ditlevsen 2016), which
can be tuned to replicate the observed signals, but since key parameters are not linked to meas-
urable quantities, these models are mostly unfalsifiable as more parameters invariably lead to
better fit, which also has limited prognostic utility since free parameters may not be assumed
fixed for a changed forcing scenario.
A more palpable paradigm is that the ice sheet is bistable (Weertman 1976), so the hys-
teresis can be triggered by eccentricity-modulated forcing to produce the 100-ky cycle, and if
the hysteresis thresholds is lowered by the putative decrease of the atmospheric $CO_2$ during the
Pleistocene, one has a possible explanation of the MPT (Berger et al. 1999; Calov and Ga-
nopolski 2005; Abe-Ouchi et al. 2013). But there is no evidence of such $CO_2$ trend (Honisch et
al. 2009), which moreover is likely a response than a cause of the climate change (Petit et al.
1999; Siegenthaler et al. 2005; Honisch et al. 2009), and then the atmospheric $CO_2$ in any event
has only minor effect on the temperature (Broecker and Denton 1989; Petit et al. 1999), the
reason that the model-produced temperature range is small compared with the observed one
(Abe-Ouchi et al. 2013). One serious difficulty of the above hysteresis is that one of the bi-
states is an ice-free state, so the ice signal is minimal prior to the MPT (Berger and Loutre





2010, Fig. 2), which is at odds with the substantial obliquity signal observed in early Pleisto-
cene (Letreguilly et al. 1991; Raymo and Nisancioglu 2003; Lisiecki and Raymo 2007). We
shall argue later (Sect. 3.3) that an ice-free state is untenable in the Pleistocene, so the hystere-
sis paradigm, if it were to apply, must involve different bistates, which as we shall see can be
engendered by the ocean.

Not sufficiently acknowledged in above studies is the central role played by the ocean.

Physically, the summer surface air temperature (SAT) that controls the ice margin is anchored
by the sea surface temperature (SST), so the ocean must constitute the primary entry of the ra-
diative forcing into the climate-cryosphere system (Broecker and Denton 1989). Observation-
ally, the meridional overturning circulation (MOC), the SST and the SAT all covary strongly
during glacial cycles (Ruddiman et al. 1986; Imbrie et al. 1992, Fig. 6; Keigwin et al. 1994,
Fig. 2; Lisiecki et al. 2008; Nie et al. 2008), which arguably precede the ice-volume signal by
several millennia, suggesting a causal linkage (Petit et al. 1999; Shackleton 2000; Medina-
Elizalde and Lea 2005). In addition, it is noted that abrupt climate changes involve MOC
jumping between modes (Broecker et al. 1985; Alley et al. 2000), a bistability that has been
demonstrated by coupled climate models (Manabe and Stouffer 1988) whose dynamical basis
however remains unclear. To remedy this shortfall, this author (Ou 2018) has recently ex-
tended Stommel's (1961) ocean-only model to include the atmospheric coupling, which reveals
bistates that depend on the forcing timescale. For sub-millennial timescale, they are the ones



seen in coarse-grain numerical models, but for orbital timescales, the bistates are set by an en-
tropy principle of the nonequilibrium thermodynamics (NT) and hysteresis can be triggered by
the modulated orbital forcing.
Justified on both physical and observational grounds, we posit therefore that the cou-
pled climate system constitutes the intermediary of the orbital forcing of the glacial cycles. Di-
ametrically opposite to numerical models that often add poorly constrained physical elements
to improve the simulations, our approach seeks to isolate minimal physics that may account for
the observed phenomenon. As organized below, our theory consists of two parts: the elucida-
tion of the inner working of a coupled climate system, and the filtering of the orbital forcing
through this system to generate glacial cycles; the two parts are contained in Sect. 2 and 3, re-
spectively. In Sect. 4, we summarize the essence of the theory and discuss how it may resolve
many Pleistocene puzzles of the glacial cycles.

## 2   Coupled climate model

In striving for minimal physics, we consider a model configuration of the North Atlan-
tic as sketched in Fig. 1, for which both ocean and atmosphere are divided into warm and cold
masses by mid-latitude fronts and an ice sheet may form on the adjacent continental strip ter-
minating at the Arctic Ocean (all symbols are defined in Appendix B). The ocean is heated by
the annual absorbed shortwave (SW) flux, which heats the overlying air by the convective and





net longwave (LW) fluxes, the latter assumed spatially uniform. Since glacial cycles are domi-
nated by the subpolar temperature, the model variables pertain to the cold-box deviations from
the global-means, the latter assumed known.
**2.1 Regime diagram**
The derivation of the model is provided in Ou (2018) to which the readers are referred
for details. Suffing for the present purpose, we shall summarize the model solution and its
underlying physics via a regime diagram shown in Fig. 2 whereby the cold-box deviations
from the global-means are plotted against the MOC ($K$). All variables have been nondimen-
sionalized (hence primed) whose scaling definitions and their standard values are listed in Ap-
pendix B. This regime diagram is drawn for a forcing of $q' = 1$ and a global convective flux
of $\bar{q}'_c = .75$.
For an infinite $K$, the ocean is homogeneous, so the cold-box SST deficit ($T'$) is zero
and the cold-box SAT ($T'_a$) is colder by the global convective flux ($\bar{q}'_c$). As $K$ decreases, the
subpolar water cools, which cools the overlying air and induces an atmospheric heat transport,
the latter in turn weakens the ocean heat transport (the total heat transport is fixed by $q'$) hence
reduces the steepness of the temperature curves. There is however an upper limit to the atmos-
pheric heat transport since the convective flux (the spacing between two temperature curves)
cannot be negative, which occurs at
$$T' = 2\bar{q}'_c. \tag{1}$$





Beyond this "convective bound", the atmospheric heat transport has saturated, and the two
temperature curves would merge to steepen at the same faster rate (inverse in $K$).

Since the moisture transport is proportional to the atmospheric heat transport on ac-

count of the Clausius-Clapeyron Equation (Ou 2007), the increasing atmospheric heat transport
with decreasing $K$ implies a salinity deficit ($S'$) that increases faster than temperature deficit $T'$
before the convective bound, but at the same rate afterward when the atmospheric heat
transport has saturated.  The disparate slopes of the two curves result in a density surplus ($\rho' =$
$T' - S'$) that has opposite slopes straddling the convective bound, the latter thus divides the cli-
mate regime into warn and cold branches, a robust outcome of the atmospheric coupling.

To specify the climate state from the continuum of these curves, one needs a constraint

on the MOC, which is customarily assumed linear in the density surplus (Stommel 1961;
Marotzke and Stone 1995) as indicated in the thick dashed line.  The proportional constant (the
inverse of the slope) is referred as the admittance and the line, the admittance line whose inter-
sects with the density curve then specify the climate state.  For the admittance chosen, the
ocean has two stable states (solid ovals) enclosing an unstable saddle point (open oval), a bista-
bility that is predicated on the convective bound hence the ocean/atmosphere coupling.

The presence of the cold state has been demonstrated by coupled numerical models

(Manabe and Stouffer 1988), which indeed is characterized by vanishing convective flux over
the subpolar water (their Fig. 17), as predicted by the convective bound.  As its further compu-
tational support, coupled models have shown hysteresis when the freshwater flux is perturbed



(Rahmstorf et al. 2005), which can be readily gleaned from the regime diagram. In these nu-
merical models, which do not resolve eddies, the admittance depends sensitively on the diapyc-
nal diffusivity --- a highly uncertain property of the ocean, which is in effect finely tuned to
replicate the observed state (Rahmstorf et al. 2005). Moreover, the hysteresis caused by chang-
ing radiative flux would be of the opposite sign of --- hence plays no part in --- the glacial cy-
cles: a stronger radiative flux, for example, would raise the temperature curve (hence lower the
density curve) to cause transition to the cold glacial instead of the warm interglacial. The cul-
prit lies in the fixed admittance line of a *laminar* ocean, which can be justified only for short-
term sub-millennial climate changes (Ou 2018), but must be relaxed for the orbital forcing of a
*turbulent* ocean, as discussed next.
**2.2 Ocean bistates**

Differing from the laminar ocean of coarse-grain numerical models, the actual MOC is

subjected to microscopic fluctuations associated with random eddy exchanges across the sub-
tropical front. Applying the fluctuation theorem (Crooks 1999), Ou (2018) deduces that the
admittance is not fixed but would evolve on millennial "entropy adjustment" time toward the
maximum entropy production (MEP). The latter thus is a veritable generalization of the sec-
ond fundamental law to nonequilibrium thermodynamics (NT, Ozawa et al. 2003), which has
been applied previously to climate theories (Kleidon 2009). We blur the admittance line by a
shaded cone to signify fluctuations, which thus would slowly pivot by increasing entropy pro-
duction until it attains maximum marked by solid ovals.



As derived in Ou (2018), the warm MEP state is given by
$$(T', K) = (q', 1/2),\qquad(2)$$
which defines our interglacial. As a crude check, applying standard parameters (Appendix B)
yields a subpolar SST of 6 $^0$C and MOC of 14 Sv (for a basin width of $6 \times 10^3 km$), not unlike
the present interglacial (Peixoto and Oort 1992; Macdonald 1998). It should be noted that
since our MOC does not depend on the uncertain diapycnal diffusivity, it is more robust than
that produced from coarse-grain numerical models.
The cold MEP on the other hand is characterized by low-salinity subpolar water at the
freezing point $T'_f$, which is consistent with the observed one during last glacial maximum
(LGM, CLIMAP 1976; Duplessy et al. 1992) hence referred as the glacial state. It is signifi-
cant that although the subpolar water is at the freezing point, it remains ice-free; this is because
the sea ice would curb the ocean heat loss to weaken the MOC hence the entropy production,
in contradiction to the MEP. This deduction however applies only for timescales long com-
pared with the millennial entropy adjustment time, so it does not preclude the extensive sea-ice
formed at the onset of the Heinrich events or during the termination of ice ages (Broecker
1994; Denton et al. 2010). Incidentally, since the sea ice has little thermal inertia hence can be
present *only* when the water is at the freezing point, it can play no active role in regulating the
glacial cycles, as previously conjectured (Gildor and Tziperman 2003).
**2.3 Hysteresis**





With the bistable MEP shown in Fig. 2, one readily discerns possible hysteresis when
the orbital forcing varies, as illustrated in Fig. 3. Suppose one is at the warm state (solid cir-
cles), a reduction in the absorbed solar flux would cool the temperature, as indicated by the
solid arrows and open circles. When the temperature cools to below the convective bound (the
horizontal dashed line marked $2\bar{q}'_c$), the warm MEP no longer exists, and the ocean would en-
ter the cold branch to propel toward the cold MEP. With (1) and (2), this cold transition $q'_c$ oc-
curs at
$$q'_c \equiv 2\bar{q}'_c, \tag{3}$$
a function only of the global convective flux, an external parameter. Now suppose one is at the
cold MEP of freezing point (solid squares), then a rising absorbed solar flux would raise the
temperature curve to propel the cold state, as indicated by dashed arrows and open squares.
The flattening of the admittance line combined with random fluctuations would vault the cli-
mate state into the warm branch, followed by its propelling toward the warm MEP. As a con-
servative upper bound on the warm transition $q'_w$ , one may set a level admittance line to yield
(Ou 2018, from his Eqs. 8 and 12)
$$q'_w \equiv (1 + \mu)\bar{q}'_c , \tag{4}$$
where $\mu$ is related to the moisture content of the atmosphere with a standard value of about .3.
The above two thresholds define the bistable interval, which thus is a function only of the
global convective flux. As the latter is decreasing during the Pleistocene cooling, its interplay



with the ice-albedo feedback and forcing modulation is seen later to produce varied glacial cy-
cles.

## 3   Glacial cycles

To translate the possibility of climate hysteresis to particulars of glacial cycles, we need

to first determine the relevant orbital forcing, as discussed next.

### 3.1 Orbital forcing

Because the thermal inertia of the ocean has integrated the seasonal cycles, the relevant

orbital forcing is the annual absorbed SW flux integrated over the subpolar water.  Being the
annual mean, the forcing is dominated by that over high latitudes, which is about an order
greater than that over the tropics (Berger et al. 2007, Fig. 2.7); and then over high latitudes, the
forcing is dominated by the summer insolation due both to the vanishing hence unvarying win-
ter insolation (Berger 1988, Fig. 18b; Tricot and Berger 1988, Fig. 4b) and to the ice-albedo
feedback that amplifies the summer forcing.  Here we must stress the necessity of the ice-al-
bedo feedback in instituting the precession forcing without which its annual absorbed flux is
zero because of the Kepler's law.  To avert this stricture, Huybers (2006) has posited a higher
but unspecified melt threshold in late Pleistocene, but with the forcing being the absorbed flux,
the ice-albedo feedback provides a palpable and quantifiable effect, as estimated next.

In early Pleistocene before the activation of the ice-albedo feedback, the forcing would

be limited to the obliquity on account of the Kepler's law.  In late Pleistocene when the ice



sheet may grow to mid-latitudes, the ice-albedo range can be as large as 0.3 (CLIMAP 1976,
Fig. 1), which would incur a range in the absorbed flux of 150 $W \cdot m^{-2}$ given the summer in-
solation of 500 $W \cdot m^{-2}$. Adding the obliquity range of 40 $W \cdot m^{-2}$ (Imbrie et al. 1993, Fig.
1), the total range is 190 $W \cdot m^{-2}$ , which would be halved to 95 $W \cdot m^{-2}$ for the annual mean.
Noting that integration over the subpolar water has allayed the latitudinal difference of the
obliquity and precession forcing, and reduced by half the excess atmospheric attenuation over
the global mean to about 10% (Tricot and Berger 1988, Fig. 1), hence both are neglected.

With the above estimates, we see that the late Pleistocene forcing is comparable to the

Milankovitch insolation (Imbrie et al. 1993, Fig. 1), which thus may be taken as its proxy. We
should stress that our use of the Milankovitch insolation is not because of its direct effect on
the summer SAT or ablation, as widely applied, but because it mimics the annual absorbed flux
that drives the SST, on which the summer SAT anchors.
**3.2 Summer SAT**

Since ice ablation is controlled by the summer SAT (Pollard 1980), the latter needs to

be linked to the annual SAT determined from our climate model (Fig. 2). The increasing conti-
nentality with latitudes induces strong latitudinal variation of both the annual and the seasonal
SAT (Oerlemans 1980; Donohoe and Battisti 2013), which on the other hand is smoothed by
the atmospheric motion. Because of these complications, it would seem intractable to deduce
the summer SAT from the heat balance, and indeed its calculation by numerical models typi-
cally involves adjusting the eddy diffusivity to replicate the observed SAT (Oerlemans 1980;



North et al. 1983). To circumvent such empirical tuning that necessarily degrades its progno-
sis, we discern nonetheless a robust constraint on the summer SAT from observations (Peixoto
and Oort 1992, Fig. 7.5), namely, it is near the freezing point at the edge of the Arctic Ocean,
which can be reasoned below on physical grounds.

We first argue that the summer air cannot be consistently warmer than the freezing

point since it would melt the perennial ice to contradict its emergence since about 3 million
year ago (Clark 1982; Ravelo et al. 2004). We then argue that the summer air may not be con-
sistently colder than the freezing point either since the Arctic Ocean would then freeze over,
which contradicts an ice-free North Atlantic on account of the MEP (Sect. 2.2) and its entry
into the Arctic Ocean that would maintain open coastal water.

At the southern end of the subpolar water, because of the much reduced continentality

hence seasonal cycle, the summer SAT can be approximated by the annual SAT determined
from our climate model. Having pinned down the two end points, we then invoke the atmos-
pheric dynamics to connect them with a straight line, as seen in Fig. 4 (the thick solid line for
the interglacial). The summer SAT would pivot about the freezing point at its northern end by
the orbital forcing (the shaded cone) whereas for the glacial state, it would assume a uniform
freezing point throughout the subpolar region hence overlay with the abscissa, a deduction that
is consistent with LGM simulations (Clark et al. 1999, Fig. 2).
**3.3 Ice margin**





Assuming a constant lapse rate, the summer SAT profile shown in Fig. 4 also repre-
sents the snowline whose intersect with the southern face of the ice sheet defines the equilib-
rium line (EL). To determine the equilibrium line altitude (ELA) $h_0$ hence the ice margin, we
need to consider both the ice dynamics and the mass balance and, given the myriad processes
involved, we shall retain only the minimal physics. For the ice dynamics, we assume a perfect
plasticity with yield stress $\tau_i$, so the momentum balance yields (see Van der Veen 2013)
$$d(h^2) = -c \cdot dx \tag{5}$$
where $h$ is the height of the southern face of the ice sheet and $c \equiv (g\rho_i)^{-1}(2\tau_i)$ with $g$, the
gravitational acceleration and $\rho_i$, the ice density. We assume an ablation rate given by $\nu T_i$
where $T_i$ is the ice surface temperature (above the freezing point) and $\nu$, an empirical constant
(Pollard 1980), and let $l_0$ and $l$ be the $x$-coordinates of the EL and ice margin, respectively,
then the annual ablation is, invoking (5),
$$A_b = \int_{l_0}^{l} \nu T_i dx$$
$$\approx \frac{\nu\gamma}{c} \int_{h_0}^{0} (h_0 - h) \, d(h^2)$$
$$= \frac{\nu\gamma}{3c} h_0^3, \tag{6}$$
which thus depends strongly on the ELA. The accumulation $A_c$ is simply the moisture flux
crossing the EL, which can be linked to the energy flux and the local moisture content --- both

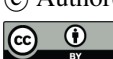



are strongly constrained by the EL being at the freezing point (Ou 2007). For this reason, the
accumulation would be largely insulated from the varying surface climate associated with the
glacial cycles and even further removed from the changing latitudinal gradient of the summer
insolation (Raymo and Nisancioglu 2003). Equating the ablation and accumulation, we derive
the ELA

$h_0 \approx (\frac{3cA_c}{\nu\gamma})^{1/3},$                                           (7)

which is estimated next.

Given the above constraint on the moisture flux by the freezing-point EL, we shall ap-

proximate it by the summer moisture flux into the Arctic Ocean as the latter is rimmed by the
freezing temperature. Based on Peixoto and Oort (1992, Fig. 12.21), this moisture flux is esti-
mated to be $A_c = 2 \times 10^5 m^2/y$. Setting additionally $\tau_i =$1 bar (Van der Veen 2013) so that
$c = 22\ m, \nu = 2\ m$ (y $^0$C) $^{-1}$ (Pollard 1980) and $\gamma = 6\ ^0C\ km^{-1}$, we estimate $h_0 = 1\ km$,
which is like that observed over the current Greenland ice sheet (Oerlemans 1991). This ELA
implies an ice margin aligned with the summer isotherm of 6 $^0$C, which may explain why the
Greenland is largely ice-covered as this isotherm is located near its southern edge (North et al.
1983, their Fig. 5b). North et al. (1983) has pegged the ice margin at the summer 0 $^0$C iso-
therm, which is deficient since there would be no summer ablation to counter the yearly accu-
mulation, and then it would imply an ice-free Greenland, contradicting the observed one. Alt-
hough the parameters in (7) are uncertain, the sensitivity is somewhat dampened by the 1/3





power, so our pegging of the ice margin with the summer isotherm of O (10 $^0$C) should gener-
ally apply.

With the above estimates, the interglacial ice sheet (the medium shade in Fig. 4) ex-

tends about 1/3 toward the subtropical front and the ice margin would vary linearly with the
forcing (the dark cone), which we shall categorize as the polar ice cap, such as the present
Greenland ice sheet.  For the glacial state, the summer SAT is at the freezing point throughout
the subpolar region, so the ice sheet would extend to the subtropical front (lightly shaded) cor-
responding to the Laurentide ice sheet (LIS).  It is seen therefore that the ocean bistates would
translate to ice sheet of vastly different sizes, resulting in a large ice-volume signal regardless
the forcing range so long as the bistable thresholds are crossed.

The large ice signal thus differs fundamentally from that associated with the inherent

ice bistates of polar ice cap and ice-free state.  And then with the surface snowline hovering
around the edge of the Arctic Ocean (Sect. 3.2), it is readily seen from Fig. 4 that the ice-free
state is unstable: a slight southward migration of the snowline would produce a finite polar ice
cap via the mass-balance feedback, yet its erasure requires a warming of about 6 $^0$C accompa-
nied by a northward snowline migration of O (1000 km) (the dashed line), an unlikely pertur-
bation.  This deduction of a stable polar ice cap is consistent with numerical studies, which
show that the current Greenland ice sheet would melt away only with more than 5 $^0$C warming
(Letreguilly et al. 1991).
**3.4 Mid-Pleistocene transition (MPT)**





The MPT from the obliquity- to the eccentricity-dominated ice-volume cycles has

posed a significant challenge to the astronomical theory since the Milankovitch insolation dis-
plays no discernible change. Besides the decreasing $CO_2$ trend that we have critiqued in Sect.
1, the changing substrate geology has also been postulated (Clark et al. 2006); both however
represent extraneous physics to the Pleistocene cooling, which is likely tectonic in origin (Rud-
diman and Raymo 1988). Instead, as a direct consequence of the Pleistocene cooling, we argue
that the global convective flux would be lowered. This is because the cooling implies a drier
air hence a smaller downward LW flux (Ou 2001, Fig. 2), which then requires a smaller global
convective flux for the ocean heat balance --- all else being equal. Significantly, the global
convective flux is precisely what sets the bistable interval of our climate, and we shall see its
interplay with the ice-albedo feedback provides a natural account of the MPT. There are of
course no proxy data for the convective flux, so its reduction during the Pleistocene cooling
can only be supported by the seeming robust physics and its potency in explaining the MPT.

Based on the discussion to follow, we show the Pleistocene evolution of the forcing and

ice signals in Fig. 5 (time proceeds to the left), which consists of three stages and their transi-
tions. The three stages correspond roughly to the three periods depicted in Imbrie et al. (1993,
their Fig. 3) and the two transitions, the early and middle Pleistocene transitions identified by
Lisiecki and Raymo (2007). For illustrative purpose, the forcing is represented by a shaded en-
velop (neglecting the obliquity and precession periods) centered on the thick dashed line (re-
ferred as the mean forcing) and the forcing markers along the right ordinate are merely indica-
tive. The vertical bars are bistable intervals spanned by the cold and warm thresholds given in



(3) and (4), which are slowly approaching the horizontal axis as the global convective flux de-
creases during the Pleistocene cooling.

At Stage 1 in the warm early Pleistocene, there is little ice-albedo feedback to effectu-

ate the precession forcing (Sect. 3.1), so the ice signal is simply that of the polar ice cap line-
arly perturbed by the obliquity forcing. This Stage 1 can be identified with the time span be-
fore 1.5 Ma (million years ago), which thus is dominated by interglacial cycles at the 41-ky
obliquity period. The continuing cooling would enhance the ice-albedo feedback hence the
precession forcing, both attaining maxima when the deepest precession trough $q'_{max}$ exceeds
the cold threshold (3) to generate the glacial state. This being the precondition of the full-
fledged precession forcing that defines Stage 2, we thus set (from Eq. (3))

$$\bar{q}'_c = q'_{max}/2 \tag{8}$$

as a crude marker for the early Pleistocene transition (EPT) from Stage 1 to 2. As the glacial
state is not yet generated prior to Stage 2, the ice signal varies linearly with the forcing, so the
EPT can be identified with the time span of 1.5-1 Ma based on the observed ice signal (Imbrie
et al. 1993, Fig. 3). Since the ice-albedo feedback primarily depresses the precession troughs
but not its peaks, the precession broadening of the forcing envelop should manifest in the deep-
ening of its mean, as indicated in the figure. This should reflect in the SST and ice volume,
which indeed is a pronounced feature in observations (Imbrie 1993, Fig. 3; McClymont et al.
2013, Fig. 8).

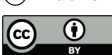



Stage 2 is defined by full-fledged precession forcing (hence modulated by eccentricity)
when the bistable centerline still lies below the mean forcing. Although glacial states are now
generated by deep precession troughs during high eccentricity, they are invariably nullified by
the next precession peaks, and the phase span of this bistate oscillation is lighter shaded to
symbolize the presence of substantial ice sheet. Outside this phase span, the precession
troughs no longer clear the cold threshold so there is only interglacial ice signal (dark-shaded)
varying linearly with the precession forcing. We shall identify Stage 2 with the time span of 1
to .7 Ma, which thus is characterized by the emerging glacial/interglacial (G/IG) cycles at the
21-ky precession period.
The continuing cooling causes the bistable centerline to rise above the mean forcing,
which defines Stage 3. There are again bistate oscillations during high eccentricity indicated
by the lighter shade, outside of which however the precession peaks no longer clear the warm
transition, so the glacial state would persist through the low eccentricity to allow the full
growth of the ice sheet to mid-latitudes. This prolonged glacial state of extensive ice sheet de-
fines the ice age, which is symbolized by the unshaded forcing envelop. Stage 3 thus is domi-
nated by ice-age cycles at the 100-ky eccentricity period, which corresponds to the observed
time span of .5 Ma to the present. It is seen from the figure that since the ice age terminates
and commences at the same warm threshold, there is no need to invoke differing physics for
their occurrences, as suspected previously (Broecker et al. 1985; Raymo 1997).
The MPT from Stage 2 to 3 thus spans the time interval of .7-.5 Ma when the bistable
centerline crosses the mean forcing, which can be seen from (3)-(4) to be given by



$$\bar{q}'_c = \frac{2}{3+\mu}\,\bar{q}^t.\tag{9}$$
While the precession is not of zero period, nor are the transitions sharply defined, which may
spread over several hundred thousand years, so the above criteria may provide crude markers
of the two transitions. For a cursory check of these criteria, we set a mean forcing of $100\ W\cdot$
$m^{-2}$ and a forcing amplitude of $50\ W\cdot m^{-2}$, then EPT and MPT would be marked by global
convective fluxes of $75\ W\cdot m^{-2}$ and $61\ W\cdot m^{-2}$, respectively. Since the global convective
flux prior to the Pleistocene should be like the present interglacial hence of O ($100\ W\cdot m^{-2}$),
and given the Pleistocene cooling of order $10\ ^{0}$C (Ruddiman and Raymo 1988, Fig. 3), the
downward LW flux as well as the global convective flux can be reduced by $50\ W\cdot m^{-2}$ (Ou
2001, Fig. 2), so the above criteria of the EPT and MPT are readily met to support their expla-
nation by the model.
The deduced three stages represent a shift of the power spectra from that dominated by
the obliquity to the emergence of the precession to that dominated by the eccentricity, as in-
deed seen in the observed ones (Imbrie et al. 1993, Fig. 3; Berger 1988, Fig. 16; McClymont et
al. 2012, Fig. 6, top panel). The emergence of the precession signal in Stage 2 would shorten
the interglacial and enhance the saw-tooth asymmetry, which are among the defining features
of the observed EPT (Lisiecki and Raymo 2007).
**3.5 Timeseries**



For a visualization of the temporal signals, we next present timeseries calculated from
the model for Stage 2 and 3 (no need to show Stage 1 characterized by interglacial signals lin-
ear in the obliquity forcing). We have argued in Sect. 3.1 that with the full operation of the
ice-albedo feedback hence precession forcing, the model forcing can be approximated by the
Milankovitch insolation, which is set to
$$q' = \bar{q}^t + \sum_{i=1}^3 a_i \cos \omega_i t. \tag{10}$$
where the time-mean forcing (the cold-box deficit) is $\bar{q}^t = 100\ W \cdot m^{-2}$, obliquity ($i = 1$) has a
period of 41 ky and amplitude $a_1 = 10\ W \cdot m^{-2}$), and precessions ($i = 2$ and $3$) have periods
of 18.5 and 23 ky with amplitudes $a_2 = a_3 = 20\ W \cdot m^{-2}$, respectively, which renders a 21 ky
precession modulated by 95 ky eccentricity. The total range of our forcing thus is $100\ W \cdot$
$m^{-2}$, as is the observed Milankovitch insolation (Berger et al. 1996, Fig. 3a).
With the above forcing, our climate model discussed in Sect. 2 would produce (time-
varying) equilibrium SST and ice margin, which are now subscripted "$e$" for distinction. To
calculate the timeseries, we apply the relaxation equations:
$$dT'/dt = (T_e' - T')/\tau_T, \tag{11}$$
and
$$dl/dt = (l_e - l)/\tau_l \tag{12}$$





where the time constant for temperature is the entropy adjustment time set at 1 ky and $\tau_l$ is the
time constant for the ice margin, which we distinguish between its advance and retreat (Weert-
man 1964). The ice advance is limited by the accumulation: for a snowfall of 0.3 m/y for ex-
ample (Ohmura and Reeh 1991), to build up an ice sheet to 3 km high takes about 10 ky, which
is thus set to be the advance time constant. The ice retreat on the other hand can be much
faster: for 2 degrees warming for example, the melt rate is 4 m/y (Pollard 1980), which is an
order greater than the accumulation, we thus set the retreat time constant to be 1 ky. The relax-
ation equations being linear, using different time constants merely affects the lag of the curves
but produces no material difference.

We show in Fig. 6 timeseries and power spectra of the forcing (solid line), the subpolar

SST (dashed) and ice margin (dotted) for Stage 2 and 3 of Fig. 5 (the MATLAB script is pro-
vided in Appendix C). The initial condition is the warm MEP state and integration is carried
forward for 400 ky; since the glacial cycles are largely repetitive, we plot only the last cycle,
the power spectra are however calculated for the full 400-ky timeseries. The upper axis repre-
sents the global-mean absorbed flux and SST. The forcing, being referenced to the former, is
expressed in its temperature equivalent ($100\ W \cdot m^{-2}$, for example, would convert to 8 $^0$C, see
Appendix B), the global-mean SST is set to $14\ ^0C$, the ice margin is its fractional extension
into the subpolar region, and the shaded bar indicates the bistable interval.

It is seen that the forcing timeseries resembles the observed Milankovitch insolation

(Berger et al. 1996, Fig. 3a) and expectedly contains no power at the eccentricity period. The
timeseries for Stage 2 (Fig. 6a) show that only one precession trough during high eccentricity





has exceeded the cold threshold to generate the glacial state characterized by freezing-point
SST and an ice sheet extending about half-way into the subpolar.  Other than this single glacial
episode lasting half the precession period, the rest of the timeseries are the interglacial SST that
tracks the forcing with slight delay and negligible ice-cap cycles.  Given the short duration of
the glacial state, the SST and ice-margin spectra show no appreciable power at the eccentricity
period, consistent with the observed spectra (Imbrie et al. 1993, Fig. 3).

The timeseries of Stage 3 differs qualitatively from that of Stage 2.  There are episodes

of interglacial during high precession peaks, which however always revert to glacial state at the
next precession trough and then there is only glacial state spanning the low eccentricity, its
long duration allows the ice sheet to grow to mid-latitudes.  Although the SST and ice-margin
spectra retain the precession and obliquity peaks as Stage 2, they show a strong eccentricity
peak absent in Stage 2.  This sharp contrast is consistent with the observed spectra (Imbrie et
al. 1993, Fig. 3).

The ice signal bears sufficient resemblance to the last ice age cycle to allow marking of

the corresponding marine isotope stages (MIS), as indicated in the figure, whose observational
features thus may be interpreted by the model physics.  According to our model, the cold sub-
stages are characterized by freezing-point subpolar water, which is consistent with the ob-
served expansion of the polar watermass and appearance of the polar species (McManus et al.
1994).  Being a glacial state, the ice growth to the mid-latitudes is only limited by the duration
of the half precession period, which has nonetheless reached half-way to the subtropical front.
This modelled ice sheet is consistent with its observational estimate (Ruddiman et al. 1980;





Chapman and Shackleton 1999), which is also supported by Ice-rafted debris (IRD) events pre-
conditioned on a large ice sheet (McManus et al. 1994). In our interpretation, all substages are
generically similar with the glacial at the cold substages reversed by the next precession peak
to the interglacial warm substages, as seen in their comparable temperature (Berger 1979, Fig.
8). It is the MIS 4 that represents the onset of the ice age (Ruddiman et al. 1980) as the suc-
ceeding precession peak fails to clear the warm threshold, resulting in prolonged coldness and
an ice sheet extending to mid-latitudes as manifested in the LIS. It is noted that the ice margin
is saw-toothed even within one precession trough due solely to the disparate advance and re-
treat rates, and this asymmetry is strongly amplified for the ice ages due to the ice growth
through the low eccentricity before the abrupt ice retreat**.**
**4   Discussion**

The central tenet of our theory is that the ocean is the intermediary of the orbital forcing

of the global ice, as strongly argued by Broecker and Denton (1989). Since the ocean is heated
by the annual absorbed flux integrated over the subpolar water, it naturally filters out latitudi-
nal difference of the obliquity and precession forcing --- except the latter would become effec-
tive when the ice-albedo feedback is activated during the Pleistocene cooling. As such, the
forcing is dominated by the obliquity component in the early Pleistocene, but can be approxi-
mated by the Milankovitch insolation in the late Pleistocene. The use of the latter in our model
thus is not because of its direct effect on the summer SAT and ablation, but because it mimics
the late-Pleistocene forcing of the ocean.



While there can be inherent bistability of finite ice sheet and ice-free state (Weertman
1961), we argue that the ice-free state is untenable in Pleistocene with the emergence of the
Arctic perennial ice about 3 Ma, as also attested by the current Greenland ice sheet and the
substantial obliquity signal even in the early Pleistocene. Rather, we posit that bistability of
the ice simply reflects that of the subpolar ocean, which in a coupled and NT climate system
may exhibit bistable warm and freezing-point temperature. Through its effect on the summer
SAT that controls the ice margin, this bistability would translate to that of the ice characterized
by polar ice cap and an ice sheet extending to mid-latitudes. The vast difference of these ice
bistates produces strong ice signal regardless the forcing perturbation so long as the bistable
thresholds are crossed.
The bistable interval of the coupled climate is linked to the global convective flux,
which would be lowered during the Pleistocene cooling that produces drier atmosphere; its in-
terplay with the ice-albedo feedback leads to three stages of the ice cycles, as well discerned in
observations (Imbrie et al. 1993; Lisiecki and Raymo 2007). In the warm early Pleistocene be-
fore appreciable ice-albedo feedback hence the precession forcing, the ice cycles are simply
that of the polar ice cap perturbed linearly by the obliquity (Stage 1). The continuing cooling
would enhance the ice-albedo feedback hence the precession forcing; while the precession
troughs during high eccentricity may induce the glacial state, it is nullified by the next preces-
sion peak, resulting in G/IG cycle at the precession period (Stage 2). With further cooling, the
precession peaks may no longer clear the warm threshold, so the glacial state would last
through the low eccentricity, resulting in ice-age cycles paced by the eccentricity (Stage 3).



The transitions between the three stages correspond to the EPT and MPT discerned in Lisiecki
and Raymo (2007), which are now assigned specific markers that can be crossed during the
Pleistocene cooling.

It should be noted that in our formulation, there are only bistable glacial and interglacial

states, the ice age is merely the glacial state that lasts through low eccentricity, there is no need
for a third full-glacial state as posited by Paillard (1998), who has not provided dynamical ba-
sis for such a state nor the transition rules among his tri-states. Since an interactive MOC is
key to the ocean bistability, numerical models that fix the SST or assume a slab ocean (North
et al. 1983; Abe-Ouchi et al. 2013) obviously cannot capture the ocean effect on the climate.
While coarse-grained coupled models have produced ocean hysteresis (Rahmstorf et al. 2005),
it is opposite in sign to that of the glacial cycle, the reason being, without resolving eddies, the
MOC is constrained by a fixed diapycnal diffusivity (Sect. 2.1). Deprived of the proper ocean
hysteresis, numerical calculations of the glacial cycles are compelled to prescribe a $CO_2$ trend
or an orbital-period $CO_2$ in augmenting the glacial signal (Berger et al. 1999; Ganopolski and
Calov 2011; Willeit et al. 2019) --- both need further justification (Sect. 1). To properly con-
strain the MOC requires resolving ocean eddies, which poses a daunting challenge to numeri-
cal models because of the long time-integration needed, but a phenomenological approach of
coding the entropy production tendency, say, via a variable eddy diffusivity, may remain feasi-
ble.
**5   Resolving glacial puzzles**





Our theory provides a single dynamic framework that may resolve seemingly unrelated
Pleistocene puzzles of the glacial cycles, as further expounded below. The reason that there is
only 41-ky obliquity cycles in the early Pleistocene is because, without the ice-albedo feed-
back, the precession has no effect on the annual absorbed flux on account of the Kepler's law,
our theory thus may resolve the "41-ky" problem. That such flux being the relevant forcing
has rendered moot some previous solutions to the 41-ky problem, such as Raymo and Ni-
sancioglu (2003) or Raymo and Huybers (2008). The MPT to the 100-ky ice-age cycle occurs
when the Pleistocene cooling has allowed the glacial state to last through the low eccentricity
hence paced by the latter; its strong signal is caused by disparate ice bistates between the polar
ice cap and an ice sheet extending to mid-latitudes; our model thus may resolve the "100-ky"
problem.
So long as the ice-age pacing is enabled by the shorter period 100-ky eccentricity, the
strength of the ice-age cycle is no longer affected by the 400-ky eccentricity even though it has
greater amplitude (Berger and Loutre 1991, Fig. 4a), the theory thus may resolve the "400-ky"
problem (Imbrie et al. 1993; Berger and Loutre 2010). In fact, it is seen from Fig. 5 that a
smaller eccentricity would produce a longer-lasting ice age to augment the ice signal, which
may explain why the 100-ky signal is gaining strength when the eccentricity is decreasing
(Clark et al. 1999, Fig. 6b) or why the lower eccentricity at Stage 11 is accompanied by higher
100-ky signal; the latter often dubbed the "Stage-11" problem (Imbrie et al. 1993, Fig. 2).
Since the onset and termination of the ice ages are threshold phenomena, both can be
off by one precession period depending on the precise timing, the ice-age cycles thus may vary





between 80- and 120-ky (Raymo et al. 1997) to resolve the "variable termination" problem.
Since the summer insolation anomalies are out-of-phase between hemispheres, their synchro-
nous glacial cycles have posed a significant puzzle (Broecker and Denton 1989), but since the
relevant orbital forcing is the annual absorbed flux, it has naturally removed this hemispheric
difference. Then with the northern ice sheet dominating the response, it would feed back onto
the global balance to synchronize the Antarctic climate, as suggested by the latter's slight lag
(a few millennia) from the Milankovitch insolation (Kawamura et al. 2007); the model thus
may possibly resolve this "polar synchronization" problem.
**Appendix A: Acronyms**
EL      Equilibrium line
ELA    Equilibrium-line altitude
EPT    Early Pleistocene transition
G/IG   Glacial/interglacial
IRD    Ice-rafted debris
Ka      Thousand years ago
Ky      Thousand years
LGM   Last glacial maximum
LIS     Laurentide ice sheet
LW     Long-wave
Ma     Million years ago
MEP   Maximum entropy production





| 553 | MIS | Marine isotope stage |
| 554 | MOC | Meridional overturning circulation |
| 555 | MPT | Mid-Pleistocene transition |
| 556 | NT | Nonequilibrium thermodynamics |
| 557 | SAT | Surface-air temperature |
| 558 | SST | Sea-surface temperature |
| 559 | SW | Short-wave |

**Appendix B: Symbols and standard values**

| 561 | $A_b$ | Ablation rate |
| 562 | $A_c$ | Accumulation rate ($= 2 \times 10^5 m^2/y$) |
| 563 | $C_{p,o}$ | Specific heat of ocean ($= 4.2 \times 10^3 J\ Kg^{-1}K^{-1}$) |
| 564 | $g$ | Gravitational acceleration ($= 9.8\ m \cdot s^{-2}$) |
| 565 | $h$ | Ice-surface height |
| 566 | $h_0$ | ELA |
| 567 | $K$ | Mass exchange rate of MOC |
| 568 | $[K]$ | Scale of $K$ ($= \alpha^* L(2\rho_o C_{p,o})^{-1} = 4.5 m^2/s$) |
| 569 | $l$ | $x$-coordinate of ice margin |
| 570 | $l_e$ | Equilibrium $l$ |
| 571 | $l_0$ | $x$-coordinate of ELA |
| 572 | $L$ | Latitudinal span of cold box ($= 3 \times 10^3 km$) |



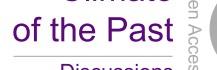

| 573 | $q'$ | Cold-box deficit of absorbed solar flux |
| 574 | $\bar{q}^t$ | Long-term mean of $q'$ $(= 100\ W \cdot m^{-2})$ |
| 575 | $[q']$ | scale of $q'$ $(= \bar{q}^t = 100\ W \cdot m^{-2})$ |
| 576 | $\bar{q}'_c$ | Global convective flux |
| 577 | $a_i$ | Amplitudes of Milankovitch insolation |
| 578 | $q'_c$ | Cold-transition threshold |
| 579 | $q'_w$ | Warm-transition threshold |
| 580 | $S'$ | Cold-box salinity deficit |
| 581 | $[S']$ | Scale of $S'$ $(= \alpha[T']/\beta = 1.79)$ |
| 582 | $\bar{T}$ | Global-mean SST $(= 14^0 C)$ |
| 583 | $T'$ | Cold-box SST deficit |
| 584 | $[T']$ | Scale of $T'$ $(= [q']/\alpha^* = 8^0 C)$ |
| 585 | $T_a'$ | Cold-box SAT deficit (from global-mean SST) |
| 586 | $T_e'$ | Equilibrium $T'$ |
| 587 | $T_f'$ | Freezing-point temperature |
| 588 | $T_i$ | Ice surface temperature |
| 589 | $\alpha$ | Thermal expansion coefficient $(= 1.7 \times 10^{-4} \cdot {}^0 C^{-1})$ |
| 590 | $\alpha^*$ | Surface transfer coefficient $(= 12.5\ W \cdot m^{-2} \cdot {}^0 C^{-1},$ Ou 2018) |
| 591 | $\beta$ | Saline contraction coefficient $(= 7.6 \times 10^{-4})$ |
| 592 | $\gamma$ | Lapse rate $(= 6^0 C/km)$ |
| 593 | $\rho'$ | Cold-box density surplus |





$[\rho']$  Scale of $\rho'$ (= $\rho_o \alpha [T'] = 1.36 \ Kg \cdot m^{-3}$)
$\rho_i$  Ice density (= $0.9 \times 10^3 \ Kg \cdot m^{-3}$)
$\rho_o$  Ocean density (= $10^3 \ Kg \cdot m^{-3}$)
$\tau_i$  Yield stress (= $1 \ bar$)
$\tau_l$  Ice-sheet time constant (= $1/10 \ ky$ for retreat/advance)
$\tau_T$  MEP-adjustment time (= $1 \ ky$)
$\mu$  Moisture-content parameter (= $0.3$)
$\nu$  Ice-melt parameter (= $1.6 \ m \cdot y^{-1} \cdot {}^0C^{-1}$)
**Appendix C:  MATLAB script**

```
% assign parameters
dt=1;tmax=400;t=(0:dt:tmax);m=length(t);m2=m/2;
fre1=2*pi/41;fre2=2*pi/18.5;fre3=2*pi/23;
amp1=0.8;amp2=1.6;amp3=1.6;
pha1=0;pha2=0;pha3=0;
dtemp1=4.5;qmean=8;dtemp2=2*dtemp1;
tempf=14;temprange=10;
tausst=1;tauac=10;tauab=1;
gamma=0.3;
% set arrays
sste=zeros(1,m);sst=zeros(1,m);
icee=zeros(1,m);ice=zeros(1,m);
qprime=zeros(1,m);q=zeros(1,m);
% initialize with interglacial state (ig)
ig=true;
for i=1:(m-1)
%insolation
qprime(i)=amp1*cos(fre1*t(i)+pha1)...
+amp2*cos(fre2*t(i)+pha2)...
+amp3*cos(fre3*t(i)+pha3);
q(i)=qmean-qprime(i);
%calculate equilibrium temperature and ice margin
if ig
sste(i)=q(i);
slt1=q(i)/2+dtemp1;
```





```
icee(i)=1-(tempf-slt1)/temprange;
if q(i)>=dtemp2
ig=false;
end
end
if ~ig
sste(i)=tempf;
icee(i)=1;
if q(i)<=(1+gamma)*dtemp1
ig=true;
end
end
640         %time integration using runge-kutta scheme
sst(1)=sste(1);ice(1)=icee(1);
sst(i+1)=runge(sste(i),sst(i),dt,tausst);
ice(i+1)=rungeice(icee(i),ice(i),dt,tauac,tauab);
end
%rescale ice margin
ice5=5*(1-ice);
ice10=10*ice;
q=tempf-q;
sst=tempf-sst;
figure
% plot timeseries
subplot(2,1,1)
plot(t,q,'-k',t,sst,'--k',t,ice5,':k')
axis([60,170,0,tempf])
set(gca,'XDir','reverse');
%legend('q','sst','ice5','location','southeast')
title({['gla6a:','amp1=',num2str(amp1),',amp2=',num2str(amp2),...
',amp3=',num2str(amp3),',pha1=',num2str(pha1),...
',pha2=',num2str(pha2),',pha3=',num2str(pha3)];...
['dtemp1=',num2str(dtemp1),',qmean=',num2str(qmean),...
',temprange=',num2str(temprange),',tauac=',num2str(tauac),',tauab=',num2str(tauab),...
',gamma=',num2str(gamma)]})
xlabel('t (ky)');ylabel('temp')
% plot power spectra
yq=fft(qprime);
yq=fftshift(yq);
sstprime=sst-mean(sst);
ysst=fft(sstprime);
ysst=fftshift(ysst);
iceprime=ice10-mean(ice10);
```





```
yice=fft(iceprime);
yice=fftshift(yice);
f=(-m/2:m/2-1)/(dt*m);
powerq=2*abs(yq).^2/(m*m);
powersst=2*abs(ysst).^2/(m*m);
powerice=2*abs(yice).^2/(m*m);
subplot(2,1,2)
plot(f,powerq,'-k',f,powersst,'--k',f,powerice,':k')
axis([0,0.1,0,3])
legend('q','sst','ice')
xlabel('freq (cycles/ky)');ylabel('power')
% runge-kutta scheme for sst
function y=runge(x,r,dt,tau)
heating=@(r) (x-r)/tau;
k1 = heating(r);
k2 = heating(r+0.5*dt*k1);
k3 = heating(r+0.5*dt*k2);
k4 = heating(r+dt*k3);
y = r+1/6*dt*(k1+2*k2+2*k3+k4);
692    % runge-kutta scheme for ice margin
function y=rungeice(x,r,dt,tauac,tauab)
msign=x-r;
if msign<0
tau=tauab;
else
tau=tauac;
end
mbalance=@(r) (x-r)/tau;
k1 = mbalance(r);
k2 = mbalance(r+0.5*dt*k1);
k3 = mbalance(r+0.5*dt*k2);
k4 = mbalance(r+dt*k3);
y = r+1/6*dt*(k1+2*k2+2*k3+k4);

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







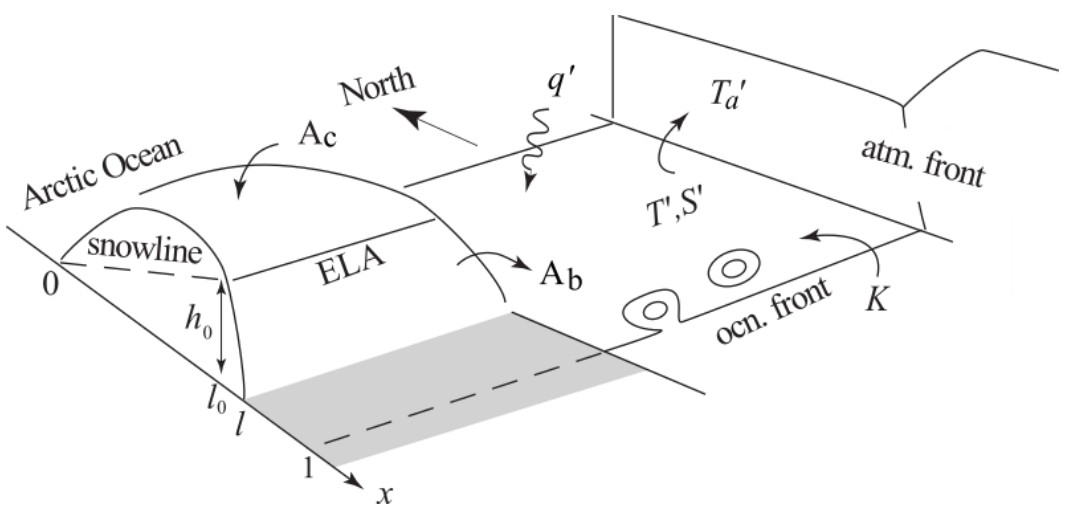


**Figure 1:** The model configuration of coupled ocean/atmosphere composed of warm and cold

boxes aligned at mi-latitudes and an ice sheet on a continental strip terminated at the Arctic

Ocean. The prognostic variables include the cold-box deviations from global means, the

MOC, and the ice margin (all symbols are listed in Appendix B)

937

938

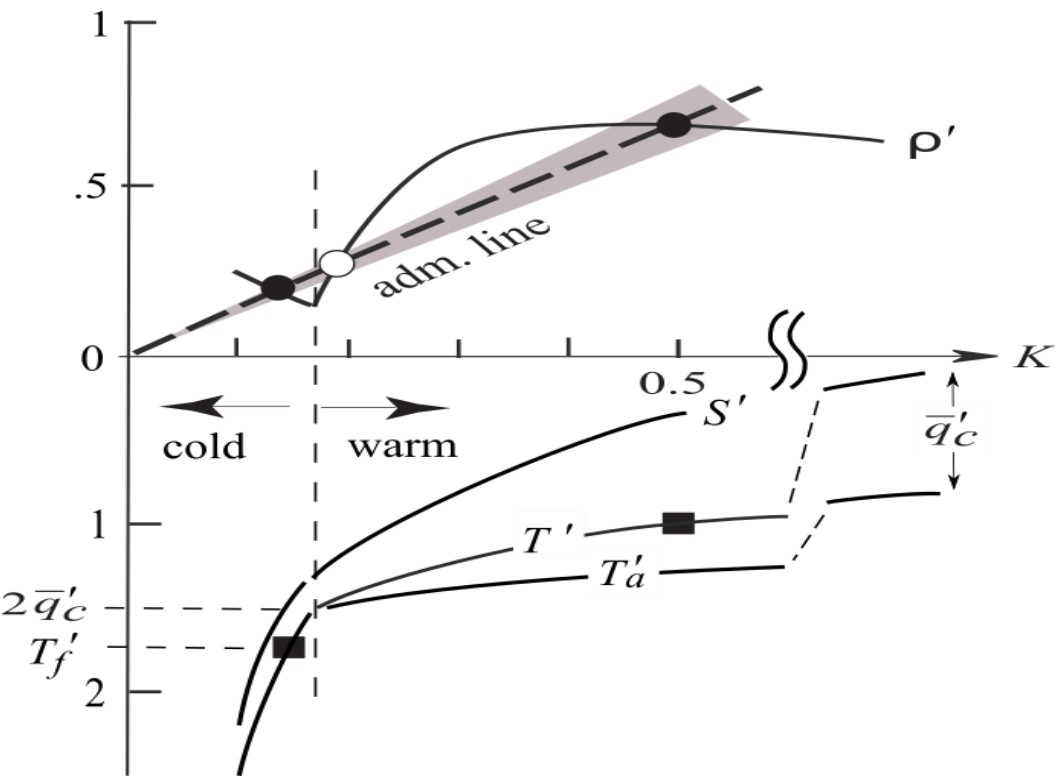

**Figure 2:** The regime diagram in which the cold-box deviations (solid lines) from the global-

means (the horizontal axis) are plotted against the MOC ($K$). The vertical dashed line marks

the convective bound when the convective flux vanishes, which divides the warm and cold

branches. The intersects of the admittance line (thick dashed) with the density curve specifies

the climate state (solid ovals). Subjected to fluctuations (shaded cone), the admittance line

would pivot toward the MEP states (solid rectangles), which define the interglacial and glacial

states, the latter characterized by the freezing-point but ice-free subpolar water. The graph is

for the case of $q' = 1$, $\bar{q}_c' = .75$, and $T_f' = 1.75$






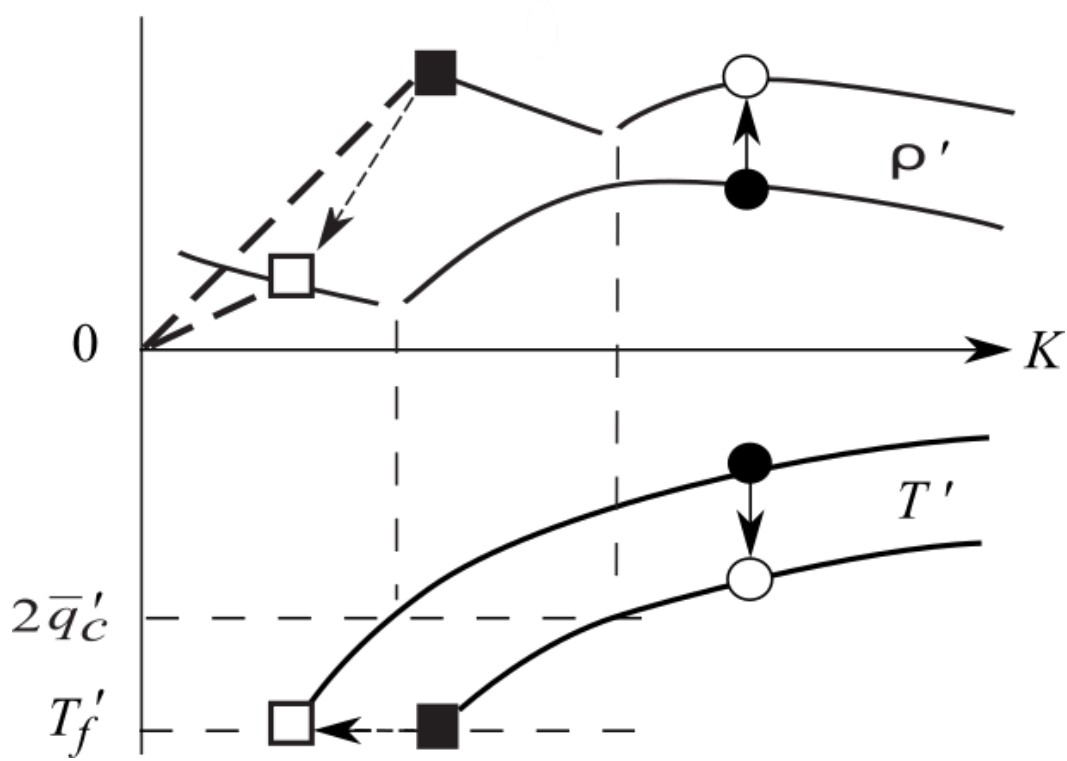


**Figure 3:** The evolution of the warm MEP when $q'$ increases (solid arrows from solid to open
circles) and the cold MEP when $q'$ decreases (dashed arrows from solid to open squares). The
thick dashed lines are the admittance lines associated with the cold MEP



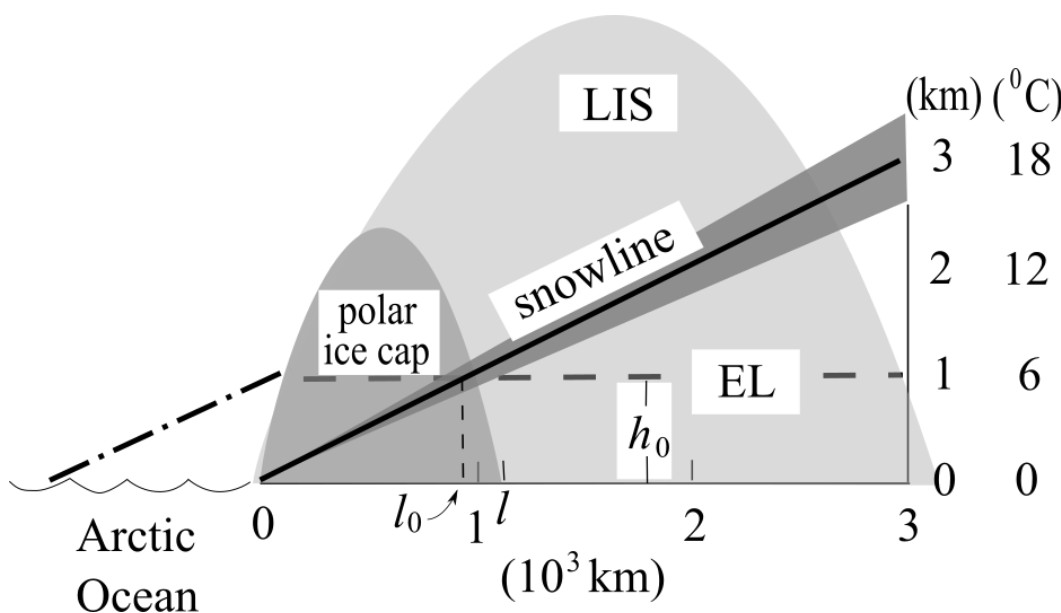


**Figure 4:** The summer SAT and snowline (aligned in the solid line) over the subpolar region


during the interglacial, the dark cone indicates their perturbation by the orbital forcing. The


ELA ($h_0$, dashed line) specifies the margin of the polar ice cap (medium shade). The dash-dot-


ted line marks the snowline when the polar ice cap may transition to the ice-free state with


strong warming. During the glacial, the summer SAT is at the freezing point (hence aligned


with the abscissa) and the ice sheet extends to the subtropical front (light shade)




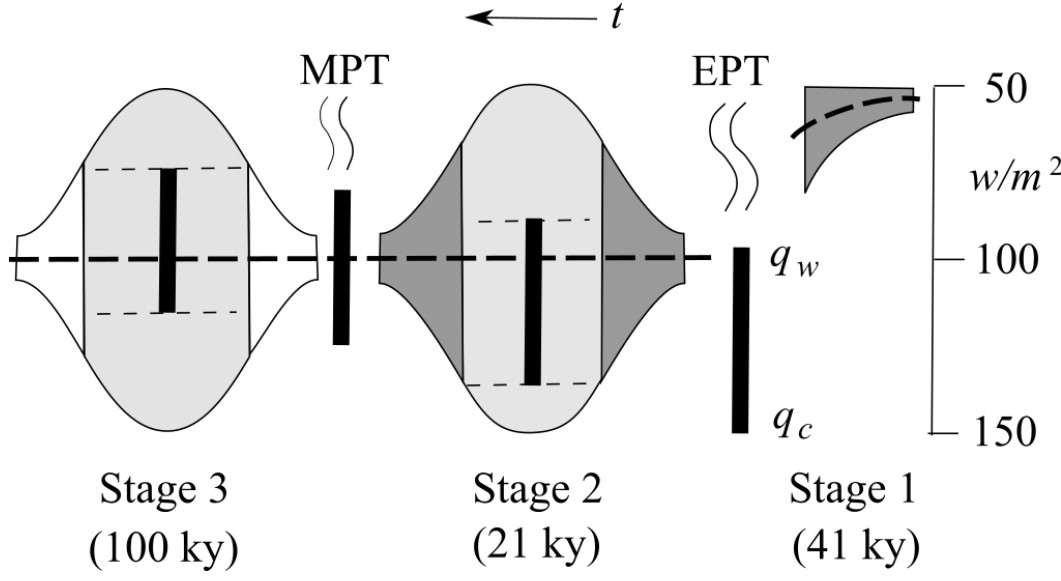


**Figure 5:** The evolution of the forcing envelop and ice signals during Pleistocene cooling,

which consists of three stages and their transitions. The vertical bars are bistable intervals

spanned by the cold ($q_c$) and warm ($q_w$) thresholds, which rise due to the Pleistocene cooling.

Stage 1 is dominated by the interglacial cycles (hence dark-shaded) at the 41-ky obliquity pe-

riod. Stage 2 sees the emergence of the G/IG cycles (hence light-shaded) at the 21-ky preces-

sion period. Stage 3 is dominated by the ice-age cycles (hence unshaded) at the 100-ky eccen-

tricity period


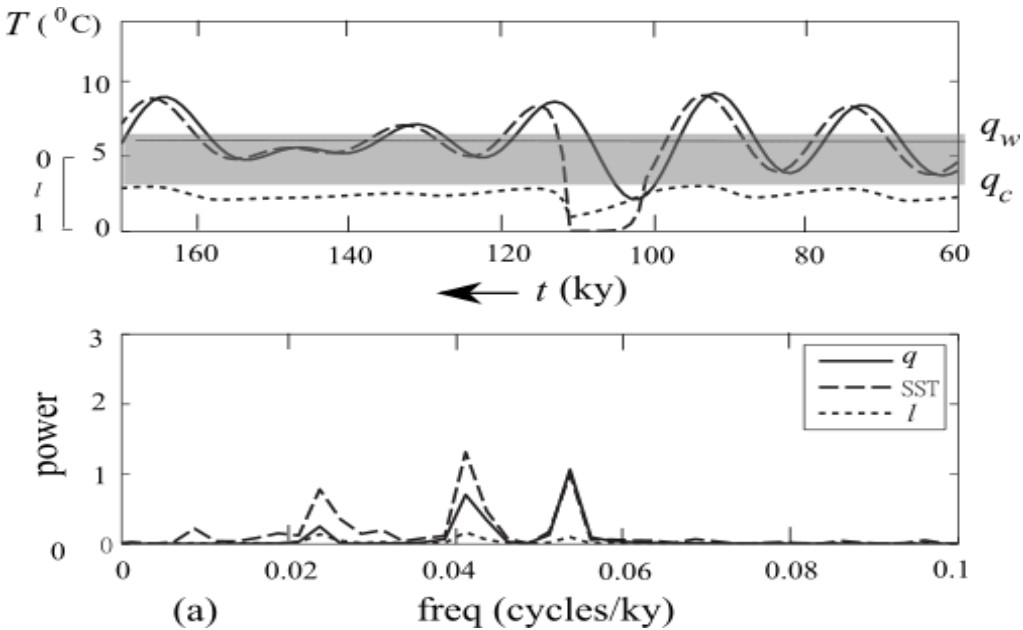


**Figure 6(a):** Timeseries and power spectra of the forcing ($q$, solid lines in equivalent tempera-
ture), subpolar SST (dashed, with a global-mean of 14 $^0$C) and ice margin ($l$, dotted, in frac-
tional extension into the subpolar). The thin horizonal line is the time-mean forcing and the
bistate interval (shaded bar) is that of Stage 2 shown in Fig. 5, which allows the generation of
the glacial state during high eccentricity, but the SST and ice-margin spectra remain dominated
by the precession





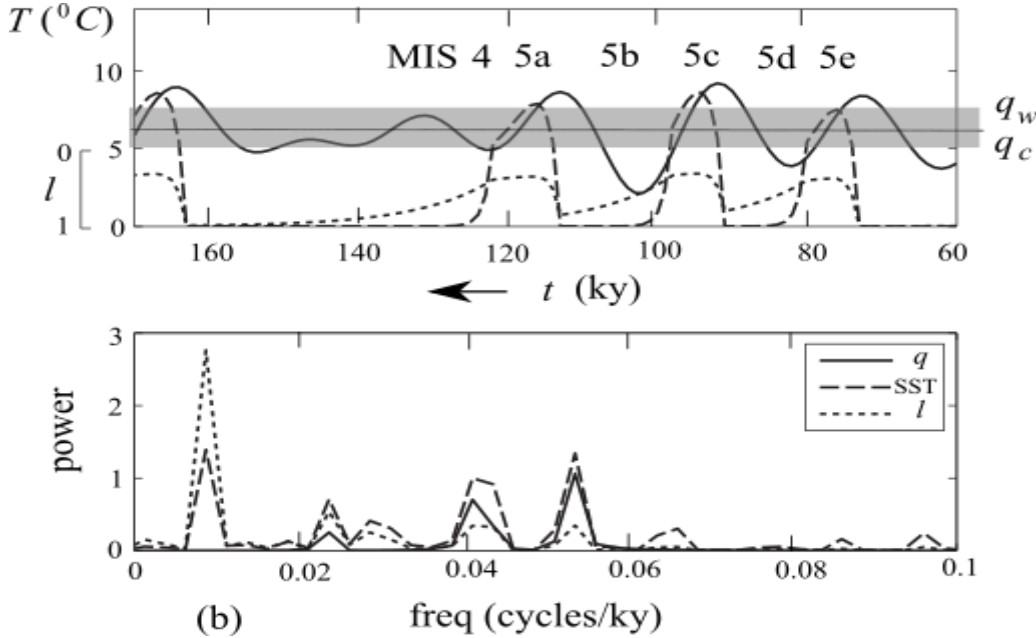


**Figure 6(b):** Same as Fig. 6a but for Stage 3 when the bistate interval (shaded bar) is further

raised by Pleistocene cooling. There are both glacial and interglacial states during high eccen-
tricity corresponding to the labelled marine isotope stages, but only the glacial state spanning
the low eccentricity, allowing the full growth of the ice sheet. The SST and ice-margin spectra
exhibit a strong peak at the eccentricity period despite its absence in the forcing spectrum
