# Peer review of "A theory of glacial cycles: resolving Pleistocene puzzles"

_Climate of the Past, 2021_

## Author Comment (AC1)

Clim. Past Discuss., referee comment RC1
https://doi.org/10.5194/cp-2021-94-RC1, 2021

[Figure]

**Comment on cp-2021-94**

Anonymous Referee #1
* * *
Referee comment on "A theory of glacial cycles: resolving Pleistocene puzzles" by
HsienWang Ou, Clim. Past Discuss., https://doi.org/10.5194/cp-2021-94-RC1, 2021
* * *
1. *I want to thank referee #1 for highly insightful comments, based on which I plan to
   overhaul the paper. In the following, I juxtapose my responses (in italics) to the referee's
   comments.*

The article claims to resolve "long-standing puzzles" with a new dynamical system model
presented as the combination of an ocean bistable system coupled with an ice
accumulation model. In essence, switches between two ocean circulation modes,
emerging by application of the maximum entropy principle, and triggered by the
astronomical forcing, control the growth and melt of continental ice. The direct response
of ice sheets to insolation changes (Milankovitch theory) is neglected: everything is
mediated by the ocean dynamics. The mid-Pleistocene transition is obtained by a
reduction of the average convective flux and the 'long-standing puzzles' resolved here are
(a) the absence of 400-ka signal (dominating eccentricity), the gain in 100-ka strength
while eccentricity decreases, a so-called 'variable termination problem' associated with
the variable length of ice age cycles, and another so-called 'polar synchronisation
problem'.

The issue at stake here is that many dynamical system models may 'resolve' these
puzzles, either with a synchronised oscillator or with non-linear resonance forced by the
astronomical forcing. For convincing us of an actual 'resolution', the present model should
have a clearly superior mechanical background compared to other models.

2. *I have provided a brief review as to why observed phenomenon has weeded out some
   previous resolutions of the glacial puzzles. One problem with the dynamical-system
   approach is that it contains free parameters that do not correspond to measurable
   quantities, so many such models are unfalsifiable (hence cannot be validated). Myriad
   mechanisms have been proposed to resolve different puzzles, the present paper in the least
   simply adds to this long list; but perhaps more significantly, it shows that by incorporating
   the well-justified ocean role and MEP, the paper may resolve all major puzzles at once. I
   venture that the quantitative derivation of the MPT and parsimony of the model physics
   represent a progress. I will further sharpen the above discussion in the revision.*

Neglecting entirely the direct ice sheet response to astronomical forcing is a provoking
proposal, because there is so much evidence of direct insolation forcing of the net ice
balance. The originality of the present setup is to use (following Ou, 2018) the maximum
entropy production principle as a better way to capture the emerging heat transport by
turbulent eddies. MEP is a fascinating but controversial topic. There is indeed a series of
articles dating from the 2000-2010 decade that suggests a good success of MEP in
predicting heat fluxes in systems with many degrees of freedom. However, some of its

main proponents, including Deware and Jupp, follow the Jaynesian interpretation of statistical mechanics: they use it more as an inference than as a prediction principle. A match between observed macro-trajectories and MEP predictions is a suggestion that the right effective constraints on the flow have been identified (see, e.g. Jupp and Cox 2010).

3. *Observations only show that the ice volume and Milankovitch insolation are correlated, not whether the linkage is direct. The only direct forcing is through the atmospheric absorption of the SW flux, as manifested in PPD, but largely overlooked is that such PPD has no precession signal because of the Kepler's law. And then whatever the summer air temperature, it is still anchored on the SST, which varies over 10 C through glacial cycles. Leaving the causality question aside, it is hard to see how SST variation does not dominate the PPD signal. I will add this discussion in the revision.*

4. *One advance of Ou (2018) is to show that MEP could be a deductive outcome of the fluctuation theorem. Since the latter is of considerable mathematical rigor and has been tested in laboratory, the MEP is more than an inference but a realizable physical state. I recently came across a DNS by Hogg and Gayen (2020), which would provide additional computational support of the MEP. Although I have reviewed MEP in Ou (2018), I plan to update the review in the revision to highlight the above points.*

5. *MEP is no longer a quirky out-of-the-mainstream idea --- considering the books, special volumes and symposia dedicated to the subject in recent years, but despite its utility in addressing the generic climate state, including that of other planets (Ou 2001; Lorenz et al. 2001), it has not entered the arena of paleoclimate research. As this paper represents arguably first such foray, it naturally would meet resistance, but I hope the transparency of the physics and its potency in resolving all major glacial puzzles would ease this resistance.*

6. *In a forthcoming paper on abrupt climate changes, I will show that MEP may explain many their salient features as well, including post-Heinrich warming, the ensuing gradual cooling that anchors D/O cycles, and the dramatic reversal of deglaciation by YD. The result should further support the utility of MEP in our understanding of the paleoclimate.*

This is important as for example, l. 170--172, the authors argue that sea-ice presence at the LGM would imply heat loss and weakened entropy production, thus "in contradiction with the MEP". Not necessarily. If it were to happen, it would not be a contradiction with MEP. It would imply that an additional constrain (here, the fact that sea-ice can actually develop) needs to be taken into account. MEP does no magic, alas !

7. *I was wrong about the sea-ice, which I realized during my current research on abrupt climate change. That is, so long as SST is hovering around the freezing point, there is invariably winter sea-ice. The MEP thus states only that for millennial or longer timescale, there can be no perennial sea ice, a deduction that is in fact consistent with LGM observations (de Vernal et al. 2005). My argument about the summer open water however still stands, and I will add the above discussion in the revision.*

Hence, my preliminary assessment is that although it is plausible that ocean dynamics have gone through some distinct states during the Pleistocene and that switches between these states have been somehow paced by the astronomical forcing, the proposal remains too speculative for claiming to have resolved "long standing puzzles". Line by line comments

8. *The new physics is that the ocean is the primary regulator of the summer air temperature, and the coupled climate would tend to MEP, both are sufficiently grounded to be deemed speculative (see responses 3 - 5). In the least, they are justifiable as working hypotheses*

*and it's their deductive outcome that resolves glacial puzzles. I will add the "working hypotheses" qualifier in the revision.*

line 8: climate system is presumably "ocean"

9. *I will retain "climate system" since a bistable ocean hinges on air-sea coupling (via the convective flux).*

line 33: 'should emerge from fundamental physics... which is yet to be delineated'. Not sure exactly what the sentence claims. Of course a model that refers to so-called fundamental principles (of physics and, perhaps, of biology ?) is to be preferred to a statistical fit. But the attempt here is not the only one to refer to such fundamental principles. Verbitsky et al. 2018 ESD claims also to provide a model rooted in fundamental principles and scaling laws, but with a focus on ice sheet dynamics and basal heat flux.

10. *I will drop "fundamental" and rephrase the sentence. I will add a reference to Verbitsky et al (2018) in the Introduction. They have a free parameter (variability number) that can be arbitrarily set to match MPT and, since eccentricity plays no role, they cannot explain why rising eccentricity paces terminations.*

ll. 57-60 : It is correct that we can't tell for sure what the pCO2 before 800 ka, but the claim of a long-term decrease in CO2 is reasonable given Pliocene proxies for CO2, even if they are very uncertain. The "no evidence" seems excessive. Atmospheric CO2 having "only a minor effect on the temperature" is a strange sentence. Yes, the radiative forcing of a 20 or 30 ppm change is small compared to the radiative effect of large ice sheet swings, but there are enough numerical simulations to claim that it is still likely to be enough to make a difference between a deglaciation terminating a 40-ka cycle, and an aborted termination that merely produces an interstadial, eventually leading to an 80-120ka cycle.

11. *I will rephrase "no evidence" to 'equivocal". A 100 ppm change in $CO_2$ only generates about 1 C temperature change (Petit et al. 1999) while observed one is 10 C. Both the carbon-cycle simulation and observed time lag suggest that $CO_2$, with its short equilibration time, is likely a response rather than a driver of the temperature signal. Although even a small $CO_2$ effect (several $Wm^{-2}$) can amplify the hysteresis (a threshold phenomenon), one may not overlook the much larger radiative effect of the drying air (several tens of $Wm^{-2}$).*

l. 79 'bistability has been demonstrated by coupled models'. Demonstrated is certainly too strong. That particular experiment by Manabe and Stouffer used the controversial freshwater flux adjustment.
Yin and Stouffer, Journal of Climate 2007, for example saw CM2.1 having no stable 'off state' (though whether two states could be obtained with a different freshwater flux background is another question). The Rahmstorf et al. 2015 is an authoritative intercomparison that remains citable today and indeed shows hysteresis for all models, but only EMICS.
This said, the recent article by Alkhayuon et al. 2019, Proceedings of the Royal Society A, presents a nice bifurcation structure for a box ocean model that may be of interest to the author.

12. *In Ou (2008, Section 4), I have provided an extensive synthesis of numerical results of the MOC hysteresis. Manabe and Stouffer's (1988) bistability indeed is a happenstance depending on the diapycnal diffusivity (hence admittance) and the strength of the air-sea coupling. Regardless the bistability however, strong enough hosing always shuts off MOC, but this off mode may persist (that is, stable) only if the initial state is bimodal, which is the source of the different stability properties found by Yin and Stouffer (2007). My reference to Manabe and Stouffer (1988) is merely to show that their off mode is indeed characterized by vanishing convective flux, hence it supports my convective bound.*

*On the other hand, the hysteresis discussed by Rahmstorf (1995) has no relevance to the glacial cycle since it is of the opposite sign: obviously a stronger insolation and warming should cause interglacial, not the cold off mode. This glacial hysteresis can only be facilitated by an admittance that is not fixed but propelled by MEP.*

13. *One problem of the ocean box model, as considered by Alkhayuon et al. (2019), is that the off mode is characterized by a reversed THC, which has no practical relevance. It is for this reason I argued in Ou (2018) that Stommel's model (1961) falls short in providing a dynamical basis for the hysteresis produced by coupled models. It is the air-sea coupling (as seen in my regime diagram) that allows a weak but normal-signed THC, which can represent a glacial ocean.*

l. 102: 'glacial cycles are dominated by the subpolar temperature' : please expand on this.

14. *I shall rephrase the sentence to "since temperature variation during the glacial cycles is dominated by subpolar over subtropical water..."*

l. 136 - 137: see above

15. *See response 12.*

l. 141: admittance could be better defined.

16. *A key element of Ou (2018) is to identify the admittance as the ocean property that is subjected to microscopic fluctuations, which is based on the observed efficacy of random eddy exchange across the subtropical front. I shall add this discussion to better define the admittance in the revision.*

l. 155 : "veritable generalization" : see introductory comments

17. *I will change "veritable" to "plausible", also see response 4.*

l. 172 : see introductory comments section 3.1 and l. 401 : Numbers are not quite right

(though order of magnitude are ok).

The range of mid-June insolation at 65N over the last million years is 435W - 559W/m2, so 123.2 W/m2.
The range of, e.g., mean insolation over the summer season (JJA, defined astronomically) is about 80W/m2.
The range of annual mean insolation at that latitude is 7 W/2m.

18. *The actual forcing is the absorbed solar flux. I use the Milankovitch insolation only as a convenient proxy hence only its crude range is needed.*

l. 302 - 304: references on the stability of the Greenland ice sheet and its future fate definitely need an update (see, e.g. Van Breedam et al. 2020, Payne et al., 2021). Clarify also whether we speak of local temperatures or global averages. There is consensus for long-term commitment to melt Greenland for globally-averaged temperatures above around 2 deg C. Is the author disputing this claim ?

19. *It's the local temperature that drives the mass balance. The required warming from Van Breedam et al. (2020) and Letreguilly et al. (1991) are not all that different: 4 C for the former and 6 C for the latter. They also are not inconsistent with the required global warming of 2 C if one applies a polar amplification factor of 2. I will rephrase the sentence to "several degrees", which does not alter my contention that an ice-free*

*Greenland is improbable during Pleistocene, as indeed attested by paleo-data. The latter
calls into question numerical simulations of the glacial cycles anchored on an ice-free
bistate. In our model, however, ice bistates simply reflect the ocean ones, which produce
more distinct ice bistates (a polar ice cap versus an ice sheet extending to mid-latitudes)
hence a stronger ice signal. I shall add above discussion and a reference to Van Breedam
et al. (2020) in the revision.*

l. 316 : if the 'cooling is tectonic in origin', what would be the mechanism ? Generally
tectonically-forced cooling implicitly refers to a tectonically-forced decrease in pCO2,
though I concede there could be other mechanisms.

20. *I have referred to Ruddiman and Raymo (1988) for a discussion of the Pleistocene
cooling, which they attribute to, among other things, the uplift that alters the albedo and
planetary waves. Not surprisingly, they have not mentioned $CO_2$, which, given its fast
equilibration, is likely a response rather than a driver of the cooling. The Pleistocene
cooling of course is a big subject with extensive literature, whose addressing lies outside
the scope of the present study; I am simply taking the cooling as an observational fact and
examine its effect on the glacial cycles. While the cooling would lower $CO_2$, its
greenhouse effect would be dwarfed by that of the drier air (several $Wm^{-2}$ versus several
tens of $Wm^{-2}$), but without incorporating the latter, the numerical calculations are
compelled to prescribe a $CO_2$ trend or its observed glacial variation in simulating the
glacial cycles, which have muddled the causality.*

l. 319 : the ocean convective flux does not need to balance changes in net IR if the
atmosphere heat flux divergence absorbs some of this change

21. *It is the global-mean convective flux, which, together with the net LW flux, balances the
absorbed SW flux; the lateral heat flux divergence plays no part in the global mean
balance. During the Pleistocene cooling, both absorbed solar flux and downward LW flux
are decreasing (the former by ice albedo and the latter by the drier air, see also Ou 2001)
while the upward LW is largely unchanged (it varies as the fourth order of the absolute
temperature), the convective flux thus must decrease. I didn't include the changing
absorbed solar flux in my original discussion, which will be added in the revision.*

l. 343 : the physical interpretation of the cause of a reduction in convective activity
remains elusive.

22. *See response 21.*

l. 369 : "differing physics" : in what sense other models require differing physics ? Clearly
the state of the ice sheet differs near full glaciation from early glaciation state, and it is
therefore natural to expect different effects of the forcing. This does not require 'differing
physics' but merely accounting for 'different states'.

23. *I will remove the sentence since it is wrong. The glaciation is smooth, occurring over
millennial entropy adjustment time, but the termination can be hastened by Heinrich events
and possibly punctuated by YD, which entail decadal abruptness (the ocean overturning
time).*

l. 421 and ll. 442 - 443: The lack of figure with a simulated time series covering the last
800 ka is disturbing.

24. *The two times series are simply to show how glacial cycles differ in Stage 2 and 3 when
global convective flux is lowered. Showing a longer timeseries with continuous global
convective flux amounts to pasting the two timeseries and smoothing their transitions,
which contains no new information.*

l. 502 : The argument would be convincing if an alternative explanation was used to justify the change in q'_c.

*25. See response 21.*

l. 496-497 : The tri-state was certainly overly schematic, but there are some explanations to the unstable character of a deeply glaciated state; glaciological interpretations evoke bedrock depletion and basal flow, and proposals giving a role to the circulation in the southern ocean / carbon cycle have also been made (Bouttes, CPast, 2012, Paillard and Parennin 2004 ,EPSL).

*26. A mode can be defined only as an attractor; different equilibria do not signify distinct modes if they vary continuously with forcing. I have not seen a dynamical basis for tri-states except quite nuanced ones discerned from numerical models, such as different convection sites or multiple ocean basins. Nor is such tri-states necessary to explain the glacial cycles, as I have demonstrated.*

l. 539 : Are Antarctic volume fluctuations driven by sea-level not a well-accepted resolution of this so-called polar synchronisation problem ? Kawamura et al. does not actually mention a 'synchronisation problem'. They made their best to accurately date terminations and confirmed indeed a northern hemisphere trigger to southern hemisphere variations.

*27. There are myriad propositions to resolve the "synchronization problem", which is premised on hemispheric anti-phase of the solar insolation. My argument is that there is no such problem in the first place if the relevant forcing is the annual absorbed SW flux. Since the ice volume signal is dominated by that of the northern hemisphere, the synchronized Antarctic signal necessarily involves global balance, which may include the sea-level.*

All that considered, it seems to me that the article makes no convincing case of a plausible alternative to the more classical approach focusing on the direct insolation forcing of nonlinear ice sheet dynamics.

*28. All studies of direct forcing link the air temperature to the Milankovitch insolation, which however overlook the fact that PPD of the direct forcing contains no precession signal because of the Kepler's law. And then the observed SST exhibits 10 C change through the glacial cycles, which would dominate PPD hence the ablation. Since such SST change hinges on an interactive MOC, models employing fixed SST or slab ocean are inadequate to capture its effect. For these reasons, numerical models that have produced realistic glacial cycles may not serve as an arbiter for the proper physics. With above responses, I hope to convince you the plausibility of the proposed physics, which in the least is justifiable as a working hypothesis.*

*29. I find your comments to be highly stimulating, which has led to much refinement in my thinking. Because of the time limit of the open discussion and the time needed for my overhaul of the paper (several weeks), I am unable to attach a revised manuscript for your perusal. I however would welcome your continuing input during this open discussion, which undoubtedly would further aid my revision effort.*

---

## Author Comment (AC4)

Clim. Past Discuss., referee comment RC2
https://doi.org/10.5194/cp-2021-94-RC2, 2021

[Figure]

**Comment on cp-2021-94**

Anonymous Referee #2
* * *
Referee comment on "A theory of glacial cycles: resolving Pleistocene puzzles" by
HsienWang Ou, Clim. Past Discuss., https://doi.org/10.5194/cp-2021-94-RC2, 2021
* * *
1.   *I want to thank the referee for the highly constructive comments, which point to significant deficiency in my presentation.  Some key points are not adequately made, which may lead to misreading of the paper.  I attach my response and planned revision below in italics.*

In this paper the authors present and elaborate on a conceptual box model of the oceanatmosphere coupled to a simple ice sheet model, where ice sheet mass balance is tied to ocean temperature. The ocean-atmosphere model is bistable and the different circulation modes result in growing or melting of the ice. Switches between different ocean states can be triggered by changes in orbital forcing.

The model is then applied to explain some of the major transitions in Quaternary climate dynamics by arguing that the global convective flux changed with Pleistocene cooling, giving rise to different model dynamics.

The argument that continental ice sheet dynamics is controlled by ocean dynamics is new and goes against the rather well-established Milankovitch theory that postulates that glacial cycles are controlled by summer insolation through its effect on ice melt. This paper still represents an interesting, although highly controversial, piece of work, but should not be sold as having solved the major 'Pleistocene puzzles'. Conceptual climate models (and this is just one of many out there) can be useful to understand particular features of a complex system but I don't think that those are the tools that will allow us to resolve all 'Pleistocene puzzles'. Physically based, spatially resolved, coupled climate-ice sheet models, whose parameters can be directly inferred from observations, are required for that.

2.   *As I stated in the first paragraph of the paper, I fully subscribe to the orbital forcing of the glacial cycles, my only difference is that this linkage is through the ocean and not through direct radiative forcing.  That the ocean plays a central role in glacial cycles is not new, but strongly argued by Broecker and Denton (1989) more than thirty years ago.  The reason that it remains overlooked could be because an interactive MOC in a turbulent ocean remains out of reach for numerical models.  I will add this point in revision.*

3.   *I agree that direct radiative forcing remains widely ascribed, but there are at least two arguments against it that seem fatal:  First, the SST ranges over 10C through glacial cycles, which incurs similar change in the SAT (basically tracking each other, see Ruddiman et al. 1986, Fig. 4; Johnsen et al. 1995, Fig. 2), the latter is greater than that can be induced by direct forcing (no more than a few degrees, see Abe-Ouchi et al. 2013, Fig. 2b).  And if the observed SAT were by direct forcing, then you have to reverse the*

*above causality – against the second law. Second, the yearly ablation from the direct forcing has no precession signal because of Kelpler's second law (Huybers 2006). Even sophisticated models that use positive degree day (for example, Abe-Ouchi et al. 2013) may not have incorporated this constraint since one cannot assume a sinusoidal seasonal insolation. I will add this discussion in revision.*

4. *I fully appreciate the value of numerical studies of glacial cycles, and a successful simulation without tuning key parameters would suggest the capture of proper physics. The present theory however has an opposite objective: instead of adding maximum physics for a realistic simulation, it seeks to isolate minimal physics sufficient to explain the observed phenomenon. The essential physics of the theory is contained in a single regime diagram (Fig. 2), and the only new element is a closure based on nonequilibrium thermodynamics necessitated by a turbulent ocean (discussed in detail in Ou 2018, including a synthesis of numerical results of the MOC hysteresis). Despite its initial guesswork more than forty years ago, MEP has entered the mainstream in recent years considering the books, special issues and symposia dedicated to the subject, so although its foray into the paleoclimate research may be new, it need not be controversial and certainly is justified as a working hypothesis. And then the potency of this principle in resolving seemingly unrelated glacial puzzles attests to its utility.*

**General comments**

Changes in the AMOC state do affect summer temperatures over North America and Europe (e.g. Jackson et al., 2015), but the magnitude of these changes is comparable or smaller than the direct changes in surface air temperature over land induced by changes in orbital configuration. Also, incoming solar radiation in summer is undoubtably important for the mass balance of an ice sheet. It therefore seems unlikely that the thermal state of the ocean alone should determine the position of the southern ice sheet margin, as assumed in the conceptual model presented in this paper. Also, there are some reconstructions of AMOC variations over glacial cycles. So the question is: should this different MOC states, implied by the conceptual model as drivers of the glacial cycles, not be reflected in proxy reconstructions of the AMOC? Is this not an 'observable' that would allow to test the model?

5. *Regardless the model result, it is an observational fact that SAT varies over 10C through glacial cycles, which is greater than that can be induced by direct forcing (see response 3). Solar insolation affects the mass balance only through summer SAT (Pollard 1980, ice absorption of SW flux is an order smaller), and this mass balance, in combination with the ice dynamics, is translated to the ELA in our model, which in turn determines the summer isotherm of the ice margin. Since summer SAT is anchored on SST, it is likely that the latter controls the ice margin. Indeed, in our theory, it is the freezing-point SST that causes LIS to extend to the subtropical front; I don't think this mid-latitude fixation of the maximum ice extent can be explained by direct forcing.*

6. *Absolutely the reconstructed MOC should provide an observational test of the model, and I have stated that SST, SAT and MOC all covary strongly, which are like that can be inferred from our regime diagram, and MOC of our warm MEP (Eq. 2) has been compared quantitatively with the observed one. On the flip side, simulations that do not allow interactive MOC or fail to reproduce its variation obviously fall short in capturing this key process.*

I find the discussion about role of CO2 for Quaternary glacial cycles on Lines 53-62 misleading and incomplete. To better understand the causes of the MPT it would be fundamental to know how CO2 changed across this transition. There are large uncertainties in CO2 reconstructions for the pre-ice core era, and a gradual CO2 decrease

over the Pleistocene is still a possible scenario (e.g. Fig.6 in Berends et al., 2021). A step forward in this respect will be represented by the planned drilling of Antarctic ice cores with ice as old as 1.5 Myr. Also, the fact that a possible long-term decrease in $CO_2$ is a consequence of the higher amplitude glacial cycles rather than the opposite, as suggested by the author, is highly speculative. The claim that $CO_2$ variations have only a minor effect on global temperature and on glacial cycles in general is not corroborated by evidence and no references are provided to support this claim. Simplified coupled climateice sheet models forced with observed $CO_2$ do produce realistic variations of global temperature between glacial and interglacial states (e.g. Fig. 8 in Ganopolski et al., 2010).

7.  *The discussion indeed is too terse, which I would expand in the revision. Berends et al. (2021) use inverse model to determine what $CO_2$ should be, but there is already observed $CO_2$, so I don't exactly follow the logic. The tectonics is the only process of long enough timescale to cause the Cenozoid cooling, and given the fast $CO_2$ equilibration, its trend through MPT (if present) or variation through the glacial cycles is likely the response than the driver of the climate change. Physically, this may be through solubility of $CO_2$ as water cools and Honisch et al. (2009) have reproduced the observed $CO_2$ of glacial cycles from SST via the carbonate chemistry. Observationally, $CO_2$ correlates strongly with SST but lags by 2 ky (Petit et al. 1999); there is no plausible reading of this other than the stated causality, which thus is far from speculative.*

8.  *$CO_2$ change of 100 ppm amounts to several $W \cdot m^{-2}$ greenhouse effect, which would be dwarfed by that of the moisture (several tens of $W \cdot m^{-2}$), so it's unclear to me why the latter is overlooked in comparison with the $CO_2$ effect (I like to be illuminated on this). Such $CO_2$ change only causes 1-2C change in SAT (Broccoli and Manabe 1987; Petit et al. 1999), which nonetheless may shift the bistable thresholds to alter the glacial cycles, as seen in numerical models. This however returns us to my primary critique about glacial cycles anchored on an ice-free state. Clearly, the observed interglacial is not ice-free, and if it were, then there will be no ice-volume signal in early Pleistocene, as seen in numerical simulations --- in contradiction to observations. Our ice bistates however are not between finite and zero ice sheet, but between polar ice cap and LIS, which have avoided the above problem.*

9.  *The foregoing problem notwithstanding, applying the $CO_2$ trend to simulate the glacial cycles may be justified by differing timescales, but imposing observed glacial $CO_2$ signal to simulate the glacial cycles strikes me as usurping the basic causality.*

The author mentions several times the need for tuning diapycnal diffusivity in ocean GCMs, and how that makes these models lose credibility. I can't see how all the assumptions and approximations made in deriving the simple model described in the paper are better than tuning a single parameter like diffusivity in an ocean general circulation model.

10.  *The problem with the sensitivity is that even within the observational bound of the diapycnal diffusivity, the MOC can be either on and off, which suggests to me that the diapycnal diffusivity is not the control parameter of the ocean state, and it emerges as such only because the models do not resolve eddies. In a turbulent ocean, it is MEP that specifies the ocean state, which has no dependence on the diapycnal diffusivity. Any theory of the scope to explain glacial cycles necessarily involve myriad assumptions, but we have tried to justify them with physical arguments when necessary.*

The long-term changes in the global convective flux play a crucial role in the explanation of the various Pleistocene climate regime shifts in the model. A more quantitative description of how this parameter is supposed to change under global cooling would be desirable. It is mentioned that it would depend on downward LW radiation, but then a

10°C (!) mean Pleistocene cooling is assumed, which, considering that temperature difference between glacials and interglacials is ~6°C, seems unreasonable. Furthermore, I imagine e.g. clouds, wind etc would also play a role.

11. *This is a valid point, and I will expand the discussion to include quantitative estimates. The global-mean surface heat balance is among the absorbed SW, the net LW and the convective fluxes. During the Pleistocene cooling, an increase of planetary albedo by 0.1 by the expanding glaciation would decrease the absorbed SW flux by 30 $W \cdot m^{-2}$, a 10C cooling would increase the net LW flux by also 30 $W \cdot m^{-2}$ (due to drying air, Ou 2001, Fig. 2), so they reinforce each other to reduce the convective flux by 60 $W \cdot m^{-2}$. This is huge, which in fact supports the robustness of its decrease even though it may not be as large. As the above estimates should at least hold in their orders of magnitude, there is no longer need for the qualification "all else being equal". Based on above estimates, the MPT markers derived in the theory may be crossed to offer a quantitative validation of our proposal.*

12. *The Pleistocene cooling is of order 10C, and superimposed on it, there is 10C range in the glacial cycles (Ruddiman and Raymo 1988, Fig. 3). The interglacial thus has similar temperature through Pleistocene, and the glacial of course is floored by the freezing point. There is nothing unreasonable about these temperature ranges. In addition, I have provided an explanation of this feature in the paragraph following Eq. (9), which is consistent with my Fig. 5.*

A continuous 3 Myr long time series of the results of the model would be interesting to see. In principle such simulation should be possible to perform by simply prescribing a scenario for the convective flux evolution over time.

13. *The timeseries are presented to provide a visual aid to the Stage 2 and 3 glacial cycles with their different convective fluxes. They are trivial products of applying the linear relaxation equation to the equilibrium state determined from Fig. 5. Producing a long timeseries amounts to juxtaposing the shown timeseries and smoothing their transition, which contains no additional information.*

**Minor comments**

Lines 221-223: sentence is not clear

14. *I will modify the sentence in revision.*

Lines 308-310: the warming threshold for Greenland melt is probably at ~2°C (Gregory et al., 2020; Robinson et al., 2012)

15. *I will modify the discussion to stress that it is an observational fact that Greenland is not ice-free either during interglacial or early Pleistocene, the only relevant point in our discussion. On the other hand, both mentioned studies are consistent with the modified statement that several degrees warming is needed for an ice-free Greenland (noting that global warming needs to be multiplied by the polar amplification factor of about 2 to get the regional temperature).*

Lines 318-320: '*This is because the cooling implies a drier air hence a smaller downward LW flux, which then requires a smaller global convective flux for the ocean heat balance --- all else being equal.*' Could the author elaborate on this? It seems far from obvious to me. How reasonable is the assumption of 'all else being equal'?

16. *See response 11.*

**References**

Berends, C. J., De Boer, B., & Van De Wal, R. S. W. (2021). Reconstructing the evolution of ice sheets, sea level, and atmospheric CO2during the past 3.6 million years. *Climate of the Past*, *17*(1), 361–377. https://doi.org/10.5194/cp-17-361-2021

Ganopolski, A., Calov, R., & Claussen, M. (2010). Simulation of the last glacial cycle with a coupled climate ice-sheet model of intermediate complexity. *Climate of the Past*, *6*(2), 229–244. https://doi.org/10.5194/cp-6-229-2010

Gregory, J., George, S., & Smith, R. (2020). Large and irreversible future decline of the Greenland ice-sheet. *The Cryosphere Discussions*, 1–28. https://doi.org/10.5194/tc-2020-89

Jackson, L. C., Kahana, R., Graham, T., Ringer, M. A., Woollings, T., Mecking, J. V., & Wood, R. A. (2015). Global and European climate impacts of a slowdown of the AMOC in a high resolution GCM. *Climate Dynamics*, *45*(11–12), 3299–3316. https://doi.org/10.1007/s00382-015-2540-2

Robinson, A., Calov, R., & Ganopolski, A. (2012). Multistability and critical thresholds of the Greenland ice sheet. *Nature Climate Change*, *2*(6), 429–432. https://doi.org/10.1038/nclimate1449

Powered by TCPDF (www.tcpdf.org)